# Influence of Sandblasting and Chemical Etching on Titanium 99.2–Dental Porcelain Bond Strength

**DOI:** 10.3390/ma15010116

**Published:** 2021-12-24

**Authors:** Malgorzata Lubas, Jaroslaw Jan Jasinski, Anna Zawada, Iwona Przerada

**Affiliations:** 1Department of Materials Engineering, Czestochowa University of Technology, Armii Krajowej 19, 42-200 Czestochowa, Poland; anna.zawada@pcz.pl (A.Z.); iwona.przerada@pcz.pl (I.P.); 2National Centre for Nuclear Research, Materials Research Laboratory, 05-400 Otwock, Poland; jaroslaw.jasinski@ncbj.gov.pl

**Keywords:** titanium 99.2, metal-ceramic system, dental porcelain, bond strength, surface treatment

## Abstract

The metal–ceramic interface requires proper surface preparation of both metal and ceramic substrates. This process is complicated by the differences in chemical bonds and physicochemical properties that characterise the two materials. However, adequate bond strength at the interface and phase composition of the titanium-bioceramics system is essential for the durability of dental implants and improving the substrates’ functional properties. In this paper, the authors present the results of a study determining the effect of mechanical and chemical surface treatment (sandblasting and etching) on the strength and quality of the titanium-low-fusing dental porcelain bond. To evaluate the strength of the metal-ceramic interface, the authors performed mechanical tests (three-point bending) according to EN ISO 9693 standard, microscopic observations (SEM-EDS), and Raman spectroscopy studies. The results showed that depending on the chemical etching medium used, different bond strength values and failure mechanisms of the metal-ceramic system were observed. The analyzed samples met the requirements of EN ISO 9693 for metal-ceramic systems and received strength values above 25 MPa. Higher joint strength was obtained for the samples after sandblasting and chemical etching compared to the samples subjected only to sandblasting.

## 1. Introduction

Metal–ceramic joints are commonly used in dental restorations, mainly due to their adequate mechanical strength and aesthetics [1,2,3]. In dental prosthetics, many substrate materials are metal alloys, which highly affect the properties of the obtained metal-porcelain joint. Platinum, gold, silver-palladium, nickel-cobalt, and titanium alloys are mainly suitable [4,5,6,7]. However, non-precious alloys are characterized by low corrosion resistance, poor biocompatibility, insufficient bonding strength, or easily formed porcelain discolouration [8,9]. 

The continuous development in biomaterials results in several studies evaluating tribological properties and the interactions between the human body and implants. In recent years, titanium and its alloys have enjoyed great success in dental applications. The continuous development in biomaterials results in several studies evaluating the interactions between the human body and titanium implants. Its main advantages include excellent biocompatibility, high strength, and corrosion resistance [10,11,12,13]. Unfortunately, such material is not without drawbacks. These include the cost of manufacturing and the complicated processing in prosthetic laboratories. Moreover, the detent to rapid surface oxidation causes some scientific papers to cite this as a primary reason for the weakening of the metal-ceramic system [14,15]. Many works have also been done on the titanium surface treatment, to obtain an accurate bonding strength of the metal-low-fusing porcelain system. Some of them concern thermal treatment, oxidation [16,17,18], modification by chemical etching [19,20,21,22,23], and laser treatment [24,25,26,27]. 

The most popular treatment of titanium surfaces used in prosthetic laboratories is sandblasting (abrasive treatment). Literature data show that the size of particles used during sandblasting is of great importance [24,28,29]. In the research performed by Walczak et al., different grain sizes of Al_2_O_3_ sand was used for the surface treatment [30]. It was shown that sandblasting particles with a diameter of 250 µm provide a higher bond strength than particles of 50 µm. Smaller diameter particles can penetrate into the metal surface (up to 30%), contaminating the substrate and weakening the bond between metal and dental porcelain. On the other hand, Al_2_O_3_ grains with a diameter of 250 µm used for sandblasting may cause an increase in surface roughness, which result in an uneven distribution of the ceramic mass on the metal substrate. As a consequence, blisters form during firing and the porcelain may chip and crack during use [31]. Based on these reports, the most favorable grain size of Al_2_O_3_ medium for sandblasting metallic surfaces is above 50 µm (optimum size 110–150 µm) [32].

Other surface treatments widely known in the prosthetics industry are chemical bathing in aqueous solutions (sodium hydroxide) and acid etching. Several studies have been carried out to improve the bonding strength of titanium–ceramic systems using procedures involving chemical modification of the surface before porcelain application [22,33]. Many authors of works dealing with this field encourage dental practices to use this type of chemical surface treatment before firing ceramics due to the better quality of the titanium-porcelain bond and, above all, the availability and simplicity of the methods [18,34,35]. However, it should be remembered that many factors influence the titanium-porcelain bond (type of porcelain, processing parameters of the metal substrate, firing conditions of the porcelain layers), which makes research on the quality of the titanium-porcelain dental bond of continued interest [36,37,38,39,40,41].

In this paper, the authors present the results of a study on the influence of mechanochemical surface treatment on the quality and strength of a low-fusing titanium-dental porcelain joint. It is hypothesised that there is a strong relationship between surface modification methods and the strength of the metal–ceramic joint. Moreover, substrate chemical etching before firing the porcelain layer can increase the bond strength of the titanium-porcelain interface. To confirm this, the authors carried out various titanium surface treatments, combining sandblasting with chemical etching in different solutions.

## 2. Materials and Methods

### 2.1. Samples Preparation

The commercially pure titanium 99.2 (KOBE Steel Group Ltd.—KS60, ASTM B265 Grade 2, Tokyo, Japan) plates were used as the substrate material. Chemical composition of the material is presented in Table 1. Plates with the dimensions of 25 (±1) mm × 3 (±1) mm × 0.5 (±1) mm were cut using water jet technique (KIMLA Waterjet StreamCut 3030, Czestochowa, Poland) and prepared according to standard EN-ISO 9693—Metal ceramic dental systems [42].

The samples were divided into three sets: Set 1—samples sandblasted with Al_2_O_3_—110 µm (reference samples), Set 2—samples sandblasted with Al_2_O_3_—110 µm and chemically etched with H_3_PO_4_ or HCl acids, Set 3—samples sandblasted with Al_2_O_3_—110 µm and chemically treated in 50% NaOH + 10% CuSO_4_ + 5H_2_O solution in combination with etching in the H_3_PO_4_ or HCl acids. Abrasive surface treatment was performed using a sandblasting device (SANDBLAST RL7-FV, Prodento-Optimed, Warsaw, Poland) at pressure of 0.25–0.30 MPa, time 10 s, sample position angle ~50° and nozzle-sample distance—10 mm. After the sandblasting and etching treatment, the samples were cleaned in distilled water using an ultrasonic cleaner for 5 min. Detailed parameters of the sandblasting and surface etching of titanium 99.2 are summarized in Table 2.

After the substrates’ surface treatment, successive layers of Duceratin Kiss (DeguDent GmbH, Hanau, Germany) porcelain were formed on the titanium samples. The thickness of the porcelain layers was approximately 1.1 mm (Bond, Opaker, Dentin). The samples were then fired according to the manufacturer’s recommendations. Firing of the Duceratin Kiss porcelain layers was carried out in accordance with the manufacturer’s instructions [43]. The view of the finished samples after heat treatment is shown in Figure 1.

### 2.2. Microstructural Analysis and Mechanical Testing

Microstructural observations were carried out together with chemical analysis in the micro areas of individual porcelain layers: Bond, Opaque, and Dentin. For the microstructural study, the scanning electron microscope (JEOL JSM-6610LV, Tokyo, Japan) equipped with an EDS analysis system was used. After the porcelain firing process, fracture force measurements were then performed, and the bond strength was determined for samples after different surface treatments. The three-point bending method was used for this purpose. Mechanical tests were carried out using a universal testing machine (Zwick/Roell GmbH & Co. KG, Ulm, Germany) in accordance with EN ISO 9693 standard [42]. The test specimens were placed with the ceramic layer downwards, as shown in Figure 2.

The measurement was carried out at a traverse feed rate of 1.5 mm/min with a load of 0.6 N. The breaking force F_fail_ was recorded as the maximum on the force-displacement curve. 

The surface morphology and chemical composition of the samples after bending tests was also analyzed with a scanning electron microscopy (SEM-EDS) and Raman spectroscopy measurements using (Horiba Jobin Yvon LabRAM HR, Kioto, Japan) spectrometer equipped with Olympus BX-41 microscope (magnification 100×), Nd: YAG 532 nm green laser, with a power set to approx. 3 mW, 2 scans with a total time of 120 s were accumulated. The obtained results finally allowed for the analysis of the surface modifications influence on the bond strength between titanium and dental porcelain.

### 2.3. Bond Strenght Estimation and Statistical Analysis

Then, using the k = f(d_M_) curve, where d_M_ is the thickness of the metal plate, the k value was read and the adhesion of the porcelain to the titanium substrate was calculated according to EN ISO 9693 [42]. In determining the bond strength (τ_b_), the following formula was used:τ_b_ = k·F_fail_(1)
where: k—coefficient depending on the thickness of the base metal and Young’s modulus,k = 4.6 (E = 113 GPa, [41]);F_fail_—metal–ceramic bond breaking force

For all types of samples, five bending strength tests were realized, and then the mean value of the destructive force (F_max_) and bending strength (τ) were determined, as well as the standard deviation (SD) and the standard error of the mean value, that is, the standard deviation of the mean value distribution (S_x_). The obtained results were statistically analysed by ANOVA analysis of variance at a 0.05 significance level. The hypothesis about the significance of differences between the mean values of flexural strength of the analysed samples was verified (hypotheses H_0_: all means are equal, H_1_: not all means are equal). Furthermore after the ANOVA test, it was necessary to verify which of the compared samples are responsible for the rejection of H_0_ and to analyse which of the means differed from each other and which were equal. For this purpose, a more thorough examination of the differences between the means of the samples was realized by multiple comparisons post-hoc test. The Fisher LSD test was used in this study and the statistics were calculated as follows:

Hypotheses:
**H_0_:** m_i_ = m_j_
**H_1_:** m_i_ ≠ m_j_
where:

i and j represent each possible combination of pairs selected from k averages
(2)$$s_{\bar{x}}=K\cdot L\cdot \sqrt{\mathrm{SK}}$$
where: K—the factor that differentiates the value of the statistic for the test,L—number associated with the number of compared groups pairs SK—intra-group variance calculated in the analysis of variance.

If $\left|{\bar{x}}_{i}-{\bar{x}}_{j}\right|\geq s_{\bar{x}};$ then reject H_0_ and accept H_1_—means in compared populations differ significantly.

If $\left|{\bar{x}}_{i}-{\bar{x}}_{j}\right|<s_{\bar{x}};$ then there are no grounds to reject H_0_—means in compared populations do not differ significantly.

## 3. Results and Discussion

The correct metal–ceramic (dental porcelain) bond depends on many factors. Some of the most important include: -surface roughness (by using blasting, unevenness is created on the metal surface, into which the porcelain penetrates, causing mechanical micro fixations) [32];-compressive stresses caused by shrinkage of metal and porcelain (thermal expansion of porcelain should be slightly smaller than metal, in which case favorable compressive stresses will occur) [29];-chemical bonding (during firing, metal oxides diffuse deep into the ceramic layers, combining with silicon oxides, and the bonds gradually change from metallic to ionic-covalent at the metal-ceramic interface) [44,45].

In order to analyze the effect of surface treatment on the metal–ceramic bond, firstly tests were carried out on individual porcelain layers. This allowed to verify the chemical composition of the layers and analyze processes occurring during the firing of the metal-ceramic system. Microscopic studies of the individual porcelain layers are shown in Figure 3a–c.

Realized SEM-EDS analysis showed a slight difference in the individual layers of dental porcelain. The primary components of all porcelain layers are Si, Zr, Al, Na, K, Ca, Ce, Ti, and O (oxides mainly represent the porcelain composition). Moreover, each layer showed the presence of a different component. In the case of Bond, it was Zr; in Opaque, Ce was additionally revealed, and in the layer of Dentin, Sn was observed. All of these components highly influence the properties of dental porcelain. Zirconium oxide is mainly used as an opacifier, but it also affects the color change and increases the fatigue resistance of the porcelain [46]. Tin oxide is introduced to lower the melting point and increase the layer hardness, while the cerium oxide increases the strength of the metal-porcelain interface by forming complex compounds such as titanites [47]. Porcelain is joined with titanium at a temperature below 800 °C. This range is suitable to prevent the formation of excessively thick oxide layers and optimal for the occurrence of a permanent and chemically stable metal-ceramic interface. The fracture force measurements and bond strength of titanium-dental porcelain systems and calculated elements necessary to determine the F test statistic are presented in Table 3 and Table 4.

The presented statistics have the Fisher-Snedecor distribution with (k − 1, *n* − k) degrees of freedom. F-statistic values greater than the critical values are the basis for the rejection of the null hypothesis (the hypothesis of equality of variance) in favor of an alternate hypothesis that indicates the presence of convergence or divergence. From the F distribution tables for the set significance level of 0.05 and for k − 1 = 4 and *n* − k = 20 degrees of freedom, the critical value F_α_ = 2.87 was read. Since F > F_α_, therefore, at the significance level of 0.05, H_0_ was rejected in favor of H_1_, which means that the average values of flexural strength differ between sample sets. Moreover, since the null hypothesis has been rejected, it is therefore necessary to verify which of the compared samples are responsible for the rejection of H_0_. It was analysed which of the means differed from each other and which were equal. For this purpose, as mentioned, multiple comparisons Fisher LSD post-hoc test was carried out. The intra-group variance calculated in the analysis of variance was SK = 17.38 and K = 2.086. The results of the Fisher LSD test are presented in Table 5.

According to EN ISO 9693 [42], dental porcelain fired on a metal substrate shall exhibit a minimum strength of 25 MPa when tested in three-point bending. When analyzing the results, it was found that all samples after mechanical and chemical surface treatment fulfilled such requirement. When comparing the results, it was found that the weakest bond between titanium and porcelain was obtained for substrates subjected to single sandblasting (Sample Set 1—30.08 MPa). Intermediate values were obtained for samples etched in H_3_PO_4_ acid (39.16 MPa) and immersed in the caustic bath NaOH + CuSO_4_ + 5H_2_O and H_3_PO_4_ acid etched (40.31 MPa). The highest values were observed for the substrates subjected to mechanochemical treatment both with etching in the 35% HCl acid (Sample Set 2) and the caustic bath NaOH + CuSO_4_ + 5H_2_O and HCl acid (Sample Set 3). The joint strength values were 42.17 MPa and 48.77 MPa, respectively. Therefore, it can be concluded that the surface etching techniques carried out for titanium substrates allowed better adhesion, and fully meet the requirements set by EN ISO 9693 standard [42]. The obtained bond strength results were higher than available literature data for similar mechanochemical surface treatments [33,48]. The authors in the research used Duceratin Kiss porcelain, which according to literature data, has higher flexural strength values than other low-fusing porcelains [49]. That is also the reason for assessing the strength of the metal–ceramic bond with regard to the type of porcelain, as its chemical composition and firing conditions may influence the value of the metal–ceramic interface strength.

After the three-point bending mechanical test, based on which the strength of the titanium-porcelain joint was determined, the surfaces (on the metal area) were subjected to morphological examination using a scanning electron microscopy. The microstructure and chemical composition analysis (SEM-EDS) results are shown in Figure 4a–e.

When analysing the surfaces, different types of damage were observed at the titanium–porcelain interface. In case of the samples from set 1 (reference sample), a complete detachment of the porcelain layers from the titanium substrate occurred according to the adhesive mechanism (all samples). SEM-EDS analysis for the mentioned variant mainly showed Ti and Al presence (Figure 4a). The presence of Al may suggest that during the sandblasting process, Al_2_O_3_ particles penetrate deep into the titanium surface. Consequently, this may affect the weakening of the titanium 99.2–Duceratin Kiss bond [50]. Analysis of the titanium surface for the sample set 2 (Al_2_O_3_/H_3_PO_4_ and Al_2_O_3_/HCl) showed the presence of porcelain components such as Zr, Si, K, Na (Figure 4b,c). It indicates that the bond fracture occurred in all of the samples at the level of the porcelain layer according to the cohesive mechanism (Figure 5C). For the samples from set 3 (Al_2_O_3_/50% NaOH + 10% CuSO_4_ + H_2_O/H_3_PO_4_ and Al_2_O_3_/50% NaOH + 10% CuSO_4_ + 5H_2_O/HCl), a mixed adhesive-cohesive type of fracture was observed and only one sample showed adhesive failure (Figure 4d,e). Despite the lower bond strength values obtained for samples etched only in acids (Sample Set 2), these substrates had a cohesive type of fracture, which is favorable from a clinical application viewpoint and most desired in metal–ceramic systems used in dentistry [51,52,53].

The differences between the titanium 99.2–porcelain interface were also investigated by Raman spectroscopy measurements (Figure 6). The chemical analysis realized for the metal–ceramics joint have shown that the samples sandblasted with Al_2_O_3_ (reference samples) showed the bands coming from the TiO_2_ rutile phase (at approx. 613 and 447 cm^−1^) (Figure 6b—band (e)) and the bands from the Ti 99.2—Duceratin Kiss porcelain interface chemical bonding (940 cm^−1^) (Figure 6a—band (a,b,c,d)) [45]. The Raman spectra obtained for TiO_2_ rutile may indicate the effect of titanium substrates oxidation during the firing process of porcelain layers. The analysis of these issues is the subject of research currently carried out by the authors and the results will be presented in the following papers. Moreover, the Raman spectroscopy results showed phase distribution change along the interface, which proved the different character of metallic and ceramic chemical bonding, which changed slightly from metallic to ionic/covalent nature.

## 4. Conclusions

The performed tests showed that chemical etching in 50% NaOH + 10% CuSO_4_ + 5H_2_O solution and H_3_PO_4_ or HCl acids improve titanium–porcelain bonding and porcelain adhesion to the titanium substrates. It should be emphasized that all of the samples met the requirements of EN ISO 9693:2019 standard [42], where the bond strength should be greater than 25 MPa. The highest bond strength value of the metal–ceramic system (48.77 MPa) was obtained for samples sandblasted with Al_2_O_3_ and immersed in a caustic bath and HCl acid etched. Furthermore, different types of titanium 99.2–porcelain Duceratin Kiss bond fracture mechanisms were observed. For sample set 2, which was sandblasted and H_3_PO_4_ or HCl acid etched, the joint was found to a cohesive fracture in the region of the ceramic layers, which is the most desired type of titanium–porcelain system failure in dentistry. In conclusion, it should be stated that realized research provides a rationale for the effective use of mechanochemical treatment processes in prosthetic laboratories to achieve favorable metal–ceramic bonds and high-quality dental implants.

## Figures and Tables

**Figure 1 materials-15-00116-f001:**
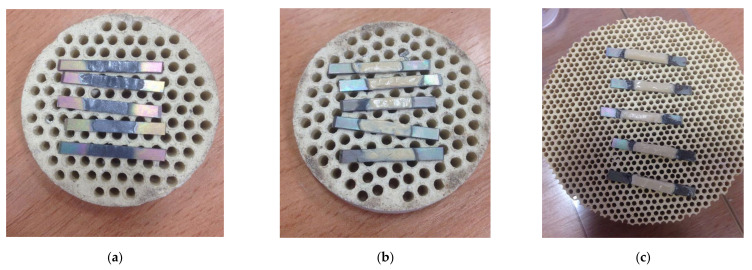
Titanium 99.2—Duceratin Kiss porcelain samples after firing process, (**a**) Bond, (**b**) Opaque, (**c**) Dentin.

**Figure 2 materials-15-00116-f002:**
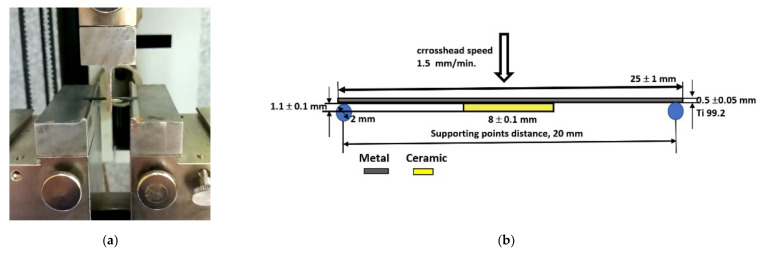
Mechanical testing of Duceratin Kiss porcelain samples (**a**) view of the samples arrangement in the bending system supports, (**b**) schematic view of the three-point bending test of a metal-ceramic system according to EN ISO 9693:2019 standard.

**Figure 3 materials-15-00116-f003:**
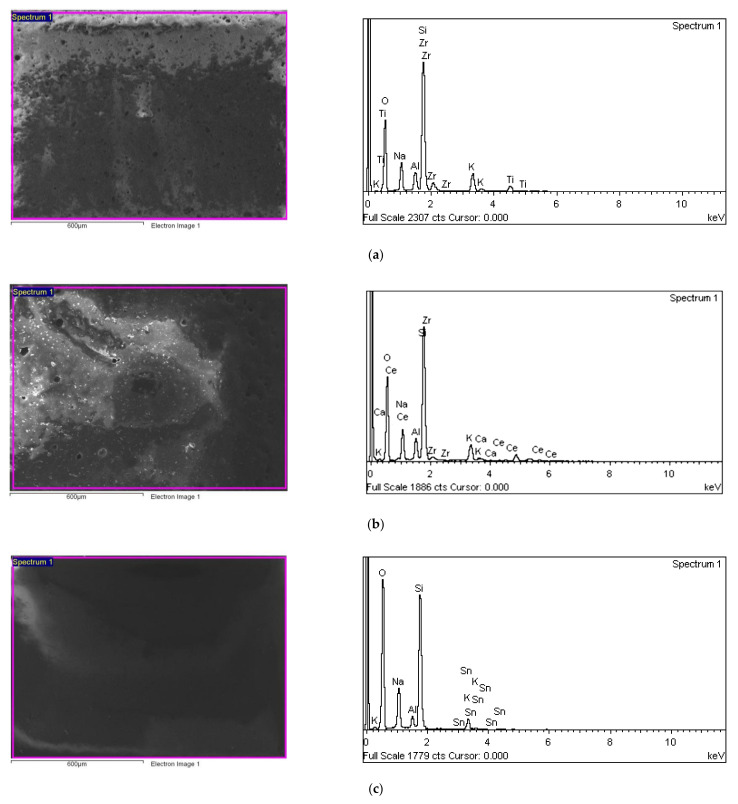
Microstructure and chemical analysis (SEM-EDS) of the substrates sandblasted with Al_2_O_3_ (reference sample) and coated with a Duceratin porcelain layers (**a**) Bond, (**b**) Opaque, (**c**) Dentin.

**Figure 4 materials-15-00116-f004:**
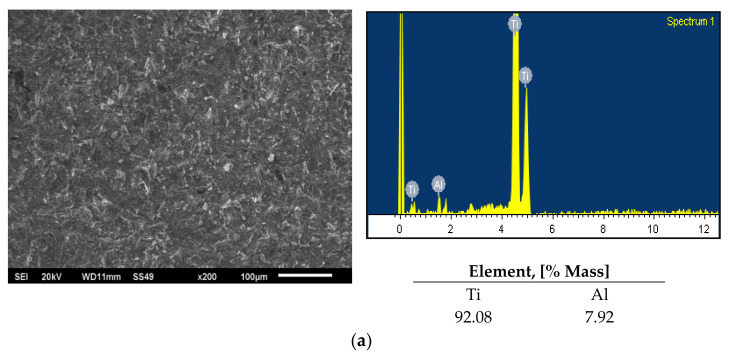
Microstructure and chemical composition analysis (SEM-EDS) of titanium 99.2−Duceratin Kiss samples after mechanochemical surface treatment, (**a**) sandblasted with Al_2_O_3_, (**b**) sandblasted with Al_2_O_3_ and H_3_PO_4_ acid etched, (**c**) sandblasted with Al_2_O_3_ and HCl acid etched, (**d**) sandblasted with Al_2_O_3_ and 10% NaOH + CuSO_4_ + 5H_2_O/H_3_PO_4_ solution etched, (**e**) sandblasted with Al_2_O_3_ and 10% NaOH + CuSO_4_ + 5H_2_O/HCl solution etched.

**Figure 5 materials-15-00116-f005:**
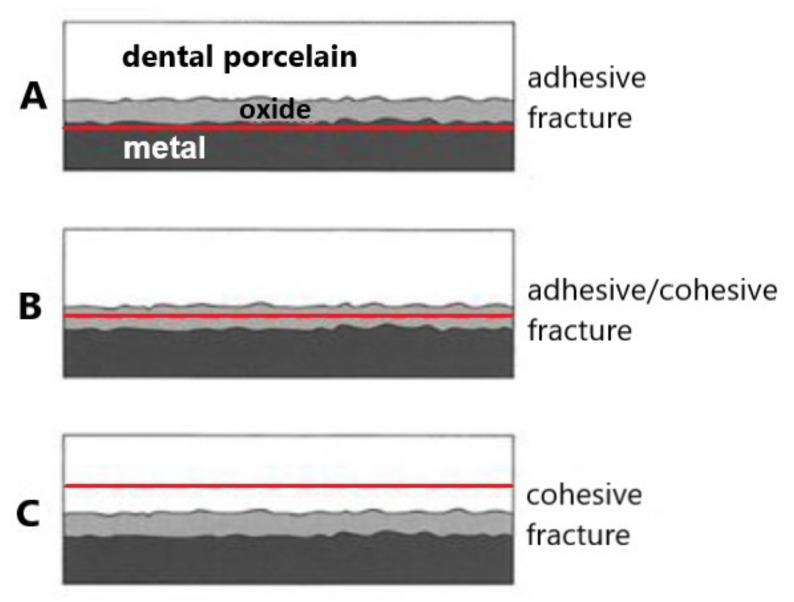
Fracture types of the Ti 99.2–Duceratin Kiss porcelain bond, (**A**) metal–oxide layer, (**B**) oxide layer–oxide layer, (**C**) porcelain layer–porcelain layer [53].

**Figure 6 materials-15-00116-f006:**
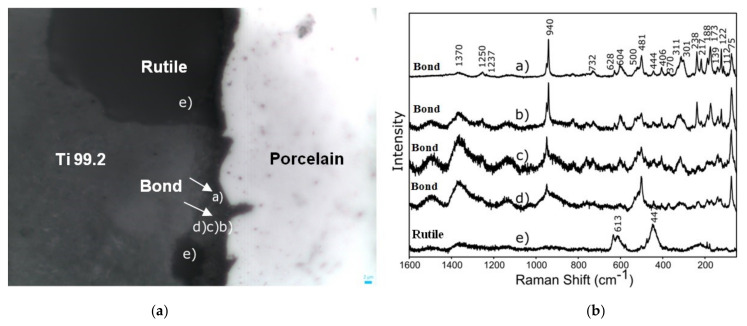
Raman spectroscopy measurements of titanium 99.2−Duceratin Kiss dental porcelain system (**a**) microscopic image of the metal–porcelain interface (**b**) Raman spectra of the bonding area.

**Table 1 materials-15-00116-t001:** Chemical composition of technically pure titanium 99.2 according to ASTM 8348.

Element, [% Mass]					
O	N	C	H	Fe	Ti
0.25	0.03	0.08	0.015	0.30	Balance

**Table 2 materials-15-00116-t002:** Variants of mechanical and chemical surface treatment of titanium 99.2 substrates.

Lp.	Surface Treatment Type	Surface Treatment Parameters
Sample Set 1		
1.	Al_2_O_3_ reference sample	1. Ultrasonic cleaning (room temp.)—5 min
		2. Sandblasting Al_2_O_3_—1 min (±5 s)
		3.Ultrasonic cleaning (room temp.)—5 min
Sample Set 2		
2.	Al_2_O_3_/H_3_PO_4_	1. Ultrasonic cleaning (room temp.)—5 min
		2. Sandblasting Al_2_O_3_—1 min (±5 s)
		3. Ultrasonic cleaning (room temp.)—5 min
		4. Etching 40% H_3_PO_4_—1 min
		5. Ultrasonic cleaning (room temp.)—5 min
3.	Al_2_O_3_/HCl	1. Ultrasonic cleaning (room temp.)—5 min
		2. Sandblasting Al_2_O_3_—1 min (±5 s)
		3. Ultrasonic cleaning (room temp.)—5 min
		4. Etching 35% HCl—1 min
		5.Ultrasonic cleaning (room temp.)—5 min
Sample Set 3		
4.	Al_2_O_3_/NaOH + 10% CuSO_4_ + 5H_2_O/H_3_PO_4_	1. Ultrasonic cleaning (room temp.)—5 min
		2. Sandblasting Al_2_O_3_—1 min (±5 s)
		3. Ultrasonic cleaning (room temp.)—5 min
		4. Etching in 50% NaOH + 10% CuSO_4_ + 5H_2_O—10 min
		5. Ultrasonic cleaning (room temp.)—5 min
		6. Etching in 40% H_3_PO_4_ acid—1 min
		7. Ultrasonic cleaning (room temp.)—5 min
5.	Al_2_O_3_/NaOH + 10% CuSO_4_ + 5H_2_O/HCl	1. Ultrasonic cleaning (room temp.)—5 min
		2. Sandblasting Al_2_O_3_—1 min (±5 s)
		3. Ultrasonic cleaning (room temp.)—5 min
		4. Etching in 50% NaOH + 10% CuSO_4_ + 5H_2_O—10 min
		5. Ultrasonic cleaning (room temp.)—5 min
		6. Etching in 35% HCl—1 min
		7. Ultrasonic cleaning (room temp.)—5 min

**Table 3 materials-15-00116-t003:** Results of mechanical tests (three-point bending) for all variants of Ti 99.2—Duceratin Kiss samples after chemical and mechanical surface treatment.

Surface Treatment Method	Fracture Type	F_max_ Fracture Force [N]	Standard Deviation SD	Standard Error of the Mean Value S_x_	τ—Bending Strength Mean Value [MPa]	Standard Deviation SD	Standard Error of the Mean Value S_x_
Sample Set 1							
Al_2_O_3_ reference sample	Adhesive	6.54	1.22	0.55	30.08	5.79	2.50
Sample Set 2							
Al_2_O_3_/H_3_PO_4_	Cohesive	8.15	1.26	0.56	39.16	2.06	0.92
Al_2_O_3_/HCl	Cohesive	9.17	1.11	0.50	42.17	5.10	0.28
Sample Set 3							
Al_2_O_3_/NaOH + CuSO_4_ + 5H_2_O/H_3_PO_4_	Adhesive/Cohesive	8.92	1.17	0.52	40.31	3.76	1.68
Al_2_O_3_/NaOH + CuSO_4_ + 5H_2_O/HCl	Adhesive/Cohesive	10.59	1.39	0.62	48.77	2.88	1.29

**Table 4 materials-15-00116-t004:** Values of the F test statistic by ANOVA analysis of variance.

**Source of Variation**	Sum of Squares of Deviations	Degrees of Freedom	Variance	Test F
Between groups	a=917.71	b =4	$\frac{a}{b}=229.43$	$F=\frac{229.43}{17.38}=13.20$
Inside groups	c =347.59	d =20	$\frac{c}{d}=17.38$	

**Table 5 materials-15-00116-t005:** Comparison of samples for significance of differences in mean flexural strength measurements by post-hoc Fisher LSD test.

Paired Samples Combination *	Differences between Means $\left|{\bar{x}}_{i}-{\bar{x}}_{j}\right|$	L	$s_{\bar{x}}$	$\left|{\bar{x}}_{i}-{\bar{x}}_{j}\right|\geq s_{\bar{x}}\;\;$
1—2	9.08	0.632	5.49	+
1—3	12.09	0.632	5.49	+
1—4	10.23	0.632	5.49	+
1—5	18.69	0.632	5.49	+

“+” the means in the populations compared are significantly different. * samples numbered according to Table 2.

## Data Availability

The data presented in this study are available on request from the corresponding author. The data are not publicly available due to possibility for use in further re-search.

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
