# Peer review of "Influence of Sandblasting and Chemical Etching on Titanium 99.2–Dental Porcelain Bond Strength"

_materials, 2021, doi:10.3390/ma15010116_

Round 1

Reviewer 1 Report

The paper is well structured, the english form is clear and accessible, the 'in vitro' study process is clearly specified.

Do you think that the sample size (unnknown, please specify) could influence the results? 

Moreover, although the max MPa have been reached in sample 3, it could be kept into account that the joint is being fractured in the area of ceramic layers, the most favorable type of destruction of the titanium-porcelain system, reached in sample 2? Please, specify if it could influence a clinical choice and why

Author Response

Dear Reviewer,

Thank you very much for your accurate comments on the article Materials 1462809 Effect of surface modification (etching and sandblasting) on the bond strength of titanium 99.2 - porcelain

Please find the attachment in .pdf (Answers to Reviewer 1 comments) where the authors answer to the Reviewer's comments.

Thank You 
Sincerely Yours,
Malgorzata Lubas 

Reviewer 2 Report

            The authors have presented a manuscript entitled “Influence of Surface Modification (Etching and Sandblasting) on the Titanium 99.2 – Porcelain Bond Strength”, where several surface treatments on Titanium 99.2 were assessed regarding bonding strength to a dental ceramics system – DeguDent’s Duceratin Kiss porcelain. Although nowadays metal-ceramic restorations are not standard of treatment for unitary or multiples fixed restorations over teeth, this system is used in the context of full-mouth implant supported rehabilitations, where there is a research interest of developing long-lasting bonds between the two materials.

Major reviews

- Since all the samples were sandblasted with aluminum oxide, the present study can’t provide new insights on mechanical treatment of titanium surfaces. This is, the hypothesis for this study stands on how the chemical titanium surface treatment influences bond strength with dental porcelain. This hypothesis should be clear at the end of the introduction section, please consider and revise accordingly.

- Regarding the methods for the present work, by the absence of a subheading for statistical analysis, I was not able to understand which data analysis was performed. Consequently, terms like “significantly” that show up in the results/discussion section can’t be related to the presented data. Please consider adding a descriptive section of the analysis performed, and the results from it.

- Still on the materials and methods, I found lacking some information on performed tests. Authors should give a concrete description of all methods performed, to better highlight the presented results, to make the work reproducible and to facilitate comparison to other published works.

Minor reviews

- Almost all sections are written in one extensive paragraph; authors should revise and make reading less dense by dividing the text.

- Page 2/Line 52 – please add reference for Walczak et al.

- Page 2/Line 63-65 – authors refer to “authors of this paper” but add references, different than those already cited before when addressing the topic in question. Please revise.

- Page 2/Line 72-74 – authors should end the introduction section with the hypothesis for the present work, not with results. Please consider and revise.

- Page 2/Line 78, Page 4/Line 109, 118, Page 6/Line 164, Page 7/Line 171, Page 10/Line 231 – several ISOs are being referenced with the same citation [42] and without a brief explanation on the ISO statement in question. Please revise.

- Page 2/Line 80-forward – the description on the samples division and treatment is quite confusing. For example, “samples etched with H3PO4 and HCl”, but this is incorrect. Samples were etched with one or another. Please revise. Also, two more sets should be created for samples, to facilitate the comparison between etching with H3PO4 and etching with HCl. Please consider.

- Along the manuscript, all materials and software must be presented along the manufacturer and country. Please revise.

- Page 2/Line 84-85 – at which distance were the titanium surfaces sandblasted? And there was an effort to sandblast evenly the titanium plate? Please revise and include it.

- Results and Discussion are advised to be separated sections by author’s guidelines provided by this journal, please revise.

- Page 5/Line 131, 132, 135 – mechanical bonding is achieved by surface roughness, compressive stress…please revise the writing. Also, add reference on the sentence ending at page 5/Line 137.

- Page 6/Line 151/152 – Add reference.

- Table 4 – authors should present the fracture types assessed on the samples and its frequency. Although three-point bending test is standard for this study, it does not represent well what biomechanically happens in the mouth. This way, the fracture types are of critical clinical importance.

- Page 7/Line 179-180 – the results from the present study are not in accordance to those in the literature. Authors should discuss why. Is this because of different methods used to assess bond strength?

- Results section – there is lacking data on statistical comparison performed. Also, a summary of the elements found at the interfaces of each sample should be presented in a clear way. Please consider.

- In terms of discussion, although the results from this study can’t directly address this question, I think authors should add some discussion regarding long-term results for the assessed bonding interfaces. Which surfaces are more prawn to resist aging, and how does it correlate with the oxides produced by different etchants?

- Conclusion should be more concise and without references. Please consider.

Author Response

Dear Reviewer,

Thank you very much for your accurate comments on the article Materials 1462809 Effect of surface modification (etching and sandblasting) on the bond strength of titanium 99.2 - porcelain

Please find the attachment in .pdf (Answers to Reviewer 2 comments) where the authors answer to the Reviewer's remarks.

Thank You 
Sincerely Yours,
Malgorzata Lubas 

Reviewer 3 Report

- This study lacks an appropriate statistical analysis (in particular, for the mechanical tests).

- The poor writing of the manuscript makes the logical progression of the paper harder to follow.

- The title should be revised for easier understanding.

- Abstract: The description about the background is too long (nearly half of the section).

- Abstract: The methods are not properly addressed.

- In the Abstract section, please avoid using subjective terms.

- How did the authors obtain the Ti plates? Please address the commercial name, if any. How did the authors prepare the plate specimens?

- According to Figure 1, the sample number for each group was 5. The, why didn’t the authors perform statistical analyses?

- The methods should be provided more structurally (using several subsection headings).

- Figs. 4 to 8: The figures should be combined in one.

- What is the results of the Raman analysis? There is no appropriate interpretation.

- The Conclusions section should be revised, only focusing on the main findings. Please avoid using subjective terms. In addition, the references (42, 51) should be omitted.

Author Response

Dear Reviewer,

Thank you very much for your accurate comments on the article Materials 1462809 Effect of surface modification (etching and sandblasting) on the bond strength of titanium 99.2 - porcelain

Please find the attachment in .pdf (Answers to Reviewer 3 comments) where the authors answer to the Reviewer's remarks.

Thank You 
Sincerely Yours,
Malgorzata Lubas 

Round 2

Reviewer 2 Report

Comments to Authors

            I thank the authors for the time invested in introducing the suggested changes. However, there are still some issues that need further attention to make this work publishable.

Major reviews

- As suggested in the first review, the hypothesis of the present work should be clearly stated at the end of the introduction section, and no results should be given in this point. The authors have not properly addressed this issue, which I think is important. Please consider.

- No comparative analysis of the results was performed. Only standard deviation and standard error of the mean were calculated. Authors should preform comparative statistical analysis, state all the procedures at a subheading of “statistical analysis” in the materials and methods section, and give its results. Only then, can authors state “significantly better adhesion” and so on if it shows up in the statistical analysis, like what’s in page 7, line 189.

- Although suggested in the first review, authors have kept Results and Discussion into one section, which does not follow author guidelines for this journal. Also, author guidelines suggest abstracts to be written into a single paragraph. Please revise.

Minor reviews

- Page 2/Line 66-67 – authors cover letter does not match what’s wrote in the manuscript, please revise.

- Still lacking some equipment detail, brands and country, for example the water jet device, the brand for the Duracetin ceramic, etc. Please revise.

- Page 7/Line 198-199 – related to the present results not being in accordance to those reported in the literature: I understand the discussion written around this subject. But in terms of writing, it shows up in different paragraphs and the discussion doesn’t seem linked to the previous sentence. Please revise writing to properly link the possibilities for the differences between the present and the already reported in the literature results.

- Page 9 – authors still did not provide frequency for fracture types, although mentioning the most common. It has to be clear if all samples revealed same fracture types or not. This is why I’ve suggested a frequency table/chart. Please consider.

- Discussion – related to what I’ve proposed in the first review “although the results from this study can’t directly address this question, I think authors should add some discussion regarding long-term results for the assessed bonding interfaces. Which surfaces are more prawn to resist aging, and how does it correlate with the oxides produced by different etchants?”, the authors have not addressed this question. Please consider.

- Discussion – additional discussion should be presented related to the concentration of the acids used. Why 40% of H3PO4 and 35% of HCl and not more or less? A brief discussion on how higher concentration may jeopardize the metal integrity leading to critical fractures or less concentrated acids can’t fully interact with metal should be provided, along the reasons behind the chosen concentrations. Please consider.

- Discussion – bond strengths were higher on the chemomechanical treated titanium surfaces, but authors have to bring together all the collected results. Although etching with NaOH+CuSO4+5H2O improved bond strengths, it does not relate with favorable fracture types. A brief discussion should be given to clarify this, and the conclusions should clearly state this different between etching alone with H3PO4 or HCl or in addition to NaOH+CuSO4+5H2O.

Author Response

Dear Reviewer,

The authors are grateful for the valuable comments and suggestions. Below are the corrections to our article in accordance with the Reviewer's comments , as well as our rationale for the changes to the article. We have revised the article and made corrections in the text. The article has also been proofread and corrected by a professional translator. The Materials and Methods chapter was divided into subsections describing the different steps of sample preparation, mechanical testing and statistical analysis. For better understanding and clarity of the article, a hypothesis was added and the description of the obtained results was revised. The authors have also corrected the bond strength values table by taking into account the fracture type of metal-ceramic substrates (Table 4) and ANOVA statistical analysis (Table 5). Conclusions to the article have also been edited to better indicate the improvement of the bond strength, confirm the hypothesis and the obtained results. We hope that the corrections made will be satisfactory to the Reviewer.

Sincerely yours,
Malgorzata Lubas

Reviewer 3 Report

The manuscript is generally well revised. However, the authors did not perform appropriate statistical analyses (for example, ANOVA and a post hoc test) for the mechanical test results. Therefore, the revised manuscript is not still acceptable.

Author Response

Dear Reviewer,

The authors are grateful for the valuable comments and suggestions. We have revised the article and made corrections in the text. The Materials and Methods chapter was divided into subsections describing the different steps of sample preparation, mechanical testing and statistical analysis. The authors have also corrected the bond strength values table by taking into account the fracture type of metal-ceramic substrates (Table 4) and ANOVA statistical analysis (Table 5). Please find attached .pdf file Answers to Reviewer's Comments.

Sincerely yours,
Malgorzata Lubas

Round 3

Reviewer 2 Report

I thank the authors for the time invested in introducing the suggested changes. However, there are still some issues that keep unanswered or were not properly addressed.

Major reviews

- The authors have based the rejection of H0 on the significant difference of the standard deviation. This does not seem the most appropriate statistical approach for the present set of data. Authors have measured bond strengths values that should be statistically compared. Analyse the distribution of the data within each set of samples and then compare the groups by its means. How do authors explain such high standard deviation (around 5.8 MPa) for the sample set 1 each corresponds to samples only sandblasted? Also, the standard error was not calculated for this set.

Minor reviews

- Statistical analysis subheading at page 5/line 134 should be place after the calculation of the bond strength since it is not considered to be statistical analysis, but how did authors assessed bond strength from the experimental design used.

- Table 4 – authors still did not present frequency of fracture types. Again, I advise you to reveal how many samples per set/experimental condition did reveal the fracture type mentioned.

- Figure 5 – authors could supplement figure 5 with the fracture types represented in a), b) and c).

Author Response

Dear Reviewer,

The authors are grateful for the valuable comments and suggestions. Following the Reviewer's remarks, the authors corrected the name of chapter as 2.3 Bond Strength Estimation and Statistical Analysis. They included the methodology for calculating the bond strength and indicated the method for performing the statistical analysis of the results obtained. The authors also to compare the variance with ANOVA analysis conducted a Fisher LSD post hoc test, the results of which we included directly in the article in red color. The authors also performed an extensive literature review of similar results, and very often, the presentation of such results did not involve such comprehensive statistical analysis. Moreover, the authors' primary intention was to present the material aspects of the chemical modification of titanium, and the article did not rely on a statistical analysis of the results. Still, in the authors' opinion, they have made a very large number of corrections. The article represents a correct approach to the research being conducted, and the results are essential to the subject of surface modification of substrates in dental applications. We hope that realized corrections made, will be satisfactory to the Reviewer.

Thank you,

Malgorzata Lubas

Reviewer 3 Report

For the mechanical test results, the authors did not perform an appropriate post hoc test (for example, Tukey).  ANOVA results do not identify which particular differences between pairs of means are significant. Post hoc tests are done to explore differences between multiple group means. According to appropriate statistical analyses including a post hoc test, the Results and Discussion section should be revised.

The Statistical Analysis subsection (2.3.) includes the calculation of the bond strengths, which has nothing to do with statistical analyses. 

Please eliminate the ANOVA table (Table 5), which is not necessary. 

Author Response

Dear Reviewer,

Following the Reviewer's remarks, the authors corrected the name of chapter 2.3 Bond Strength Estimation and Statistical Analysis. They included the methodology for calculating the bond strength and indicated the method for performing the statistical analysis of the results obtained. The authors also performed an ANOVA analysis of variance according to the Reviewer's earlier remarks, which shows that the substrates can be compared to each other. We also conducted a Fisher’s LSD  post hoc test, the results of which we included directly in the article at the Reviewer's request. Unfortunately, we did not remove the table with the ANOVA test because this change was accepted by Reviewer 2. The authors also performed an extensive literature review of similar results, and very often, the presentation of such results did not involve such comprehensive statistical analysis. Moreover, the authors' primary intention was to present the material aspects of the chemical modification of titanium, and the article did not rely on a statistical analysis of the results. However, the authors, have done a complete statistical analysis of the results obtained, including the test of variance, but this still does not satisfy the Reviewer. Initially, the Reviewer expected only the ANOVA test, which the authors included in the text, but now the Reviewer expects a post hoc test. In the authors' opinion, they have made a very large number of corrections and the article represents a correct approach to the research being conducted. We think that the results in the article are essential to the subject of surface modification of substrates in dental applications and we hope that realized corrections made, will be satisfactory to the Reviewer.

Please find .pdf file with the Answers to Reviewer remarks

Thank you,
Malgorzata Lubas
